# Preparation and Properties of Intrinsically Atomic-Oxygen Resistant Polyimide Films Containing Polyhedral Oligomeric Silsesquioxane (POSS) in the Side Chains

**DOI:** 10.3390/polym12122865

**Published:** 2020-11-30

**Authors:** Hao Wu, Yan Zhang, Yi-Dan Guo, Hao-Ran Qi, Yuan-Cheng An, Yan-Jiang Jia, Yao-Yao Tan, Jin-Gang Liu, Bo-Han Wu

**Affiliations:** 1Beijing Key Laboratory of Materials Utilization of Nonmetallic Minerals and Solid Wastes, National Laboratory of Mineral Materials, School of Materials Science and Technology, China University of Geosciences, Beijing 100083, China; 2003200021@cugb.edu.cn (H.W.); 3003200016@cugb.edu.cn (Y.Z.); guoyidan163@163.com (Y.-D.G.); 2103200030@cugb.edu.cn (H.-R.Q.); 2103190039@cugb.edu.cn (Y.-C.A.); 2003190024@cugb.edu.cn (Y.-J.J.); 2003190022@cugb.edu.cn (Y.-Y.T.); 2Space Materials and Structure Protection Division, Beijing Institute of Spacecraft Environment Engineering, Beijing 100094, China

**Keywords:** polyimide film, atomic oxygen, POSS, thermal properties, tensile properties

## Abstract

The relatively poor atomic-oxygen (AO) resistance of the standard polyimide (PI) films greatly limits the wide applications in low earth orbit (LEO) environments. The introduction of polyhedral oligomeric silsesquioxane (POSS) units into the molecular structures of the PI films has been proven to be an effective procedure for enhancing the AO resistance of the PI films. In the current work, a series of POSS-substituted poly (pyromellitic anhydride-4,4′-oxydianiline) (PMDA-ODA) films (POSS-PI) with different POSS contents were synthesized via a POSS-containing diamine, *N*-[(heptaisobutyl-POSS)propyl]-3,5-diaminobenzamide (DABA-POSS). Subsequently, the effects of the molecular structures on the thermal, tensile, optical, and especially the AO-erosion behaviors of the POSS-PI films were investigated. The incorporation of the latent POSS substituents decreased the thermal stability and the high-temperature dimensional stability of the pristine PI-0 (PMDA-ODA) film. For instance, the PI-30 film with the DABA-POSS content of 30 wt% in the film exhibited a 5% weight loss temperature (*T*_5%_) of 512 °C and a coefficient of linear thermal expansion (CTE) of 54.6 × 10^−6^/K in the temperature range of 50–250 °C, respectively, which were all inferior to those of the PI-0 film (*T*_5%_ = 574 °C; CTE = 28.9 × 10^−6^/K). In addition, the tensile properties of the POSS-containing PI films were also deteriorated, to some extent, due to the incorporation of the DABA-POSS components. The tensile strength (*T*_S_) of the POSS-PI films decreased with the order of PI-0 > PI-10 > PI-15 > PI-20 > PI-25 > PI-30, and so did the tensile modulus (*T*_M_) and the elongations at break (*E*_b_). PI-30 showed the *T*_S_, *T*_M_, and *E*_b_ values of 75.0 MPa, 1.55 GPa, and 16.1%, respectively, which were all lower than those of the PI-0 film (*T*_S_ = 131.0 MPa, *T*_M_ = 1.88 GPa, *E*_b_ = 73.2%). Nevertheless, the incorporation of POSS components obviously increased the AO resistance of the PI films. All of the POSS-PI films survived from the AO exposure with the total fluence of 2.16 × 10^21^ atoms/cm^2^, while PI-0 was totally eroded under the same circumstance. The PI-30 film showed an AO erosion yield (*E_s_*) of 1.1 × 10^−25^ cm^3^/atom, which was approximately 3.67% of the PI-0 film (*E_s_* = 3.0 × 10^−24^ cm^3^/atom). Inert silica or silicate passivation layers were detected on the surface of the POSS-PI films after AO exposure, which efficiently prevented the further erosion of the under-layer materials.

## 1. Introduction

Standard wholly aromatic polyimide (PI) films, such as poly(pyromellitic anhydride-4,4′-oxy- dianiline) (PMDA-ODA, trademark: Kapton^®^ by DuPont, USA) films, have been widely used in modern industry for more than half a century since the commercialization in 1960 s [1,2,3]. The PI (PMDA-ODA) films are usually known for the excellent combined properties, including the wide servicing temperature range from −269 °C to 300 °C, excellent mechanical and dielectric properties, good environmental stability, and so on; thus, can usually meet most of the property requirements in high-tech fields. However, the standard PI (PMDA-ODA) film has been facing increasing challenges with the expanding applications in aeronautical, optoelectronic, and other specific areas in the past decades. For example, PI (PMDA-ODA) film is easily eroded by the atomic oxygen (AO) explosion, which is one of the most important space environments in low earth orbit (LEO) [4]. The PI (PMDA-ODA) film has been thought to be good candidate as the protecting components for the LEO spacecrafts due to the excellent combined properties. However, the in-orbit experimental results performed in the LEO spacecrafts indicated that obvious mechanical property deterioration and weight loss occurred in the standard PI (PMDA-ODA) films, due to the high energy up to 5 eV during the impact with the AO species at the cruising speed of 7.8 km/s [5,6,7,8,9,10]. This energy is usually higher than the dissociation energy of the chemical bonds in the PI (PMDA-ODA) films, such as C-N (3.2 eV), C-C (3.9 eV), and so on [11]. Thus, the enhancement of AO-resistant property of PI (PMDA-ODA) films has attracted increasing attention from both the academic and engineering aspects.

Up to now, there have been two procedures for enhancing the AO resistance of the PI (PMDA-ODA) films, which are known as the passive protection and active protection. The former procedure is to incorporate the AO-resistant components, mainly inorganic or metal oxides, such as silica, alumina, zirconia, and so on, into the matrix or onto the surface of the pristine PI (PMDA-ODA) films [12,13,14,15]. These specific components could react with the AO species to afford the AO-resistant passivation layers onto the surface of PI film; thus, providing the long-term AO protection. Although this procedure is usually highly efficient and cost-effective, it is often subject to the deterioration of the optical or mechanical properties of the composite films, due to the compatibility of the fillers with the matrix or the uniformity of the coatings. In addition, the “undercutting” phenomena can usually be observed for the surface-passivated PI films, due to the undetectable pinholes or defects in the AO-resistant coatings [16]. The latter procedure was paid considerable attention in recent years, by which some specific elements, such as silicon and phosphorus, were first incorporated into the dianhydride or diamine monomers, and then introduced into the molecular structure of the PI films by polymerization of the specific monomers. The active components can react with AO to in-situ form the passivation layer. Thus, the PI films themselves possess the AO-resistant ability and exhibit the “self-passivating” or “self-healing” features in the AO environments [17,18]. Thompson et al. reported the PI films that were prepared from a phosphorus- containing dianhydride, 4,4′-(2-diphenylphosphinyl-1,4-phenylenedioxy)- diphthalic anhydride for potential space applications [19]. The prepared PI films showed low solar absorptivity, good mechanical properties, and good AO resistance. Watson et al. reported the space environmentally stable PI and copolyimide films that were derived from a phosphorus-containing diamine, [2,4-bis(3-aminophenoxy)phenyl]diphenylphosphine oxide [20]. The derived PI films exhibited good AO resistance, together with the low solar absorptivity, high optical transparency, and good tensile properties. Li and coworkers reported the atomic oxygen-resistant and transparent PI films from [3,5-bis(3-aminophenoxy)phenyl] diphenylphosphine oxide and aromatic dianhydrides [21]. The developed PI film showed good optical transparency, low yellowness, low refractive indices, and high stability in the AO environments with the erosion yield of 6.59 × 10^−25^ cm^3^/atom.

Besides the phosphorus-containing groups, silicon-containing substituents have also been used for developing intrinsically AO-resistant PI films. Either the linear siloxane linkages or the cyclic polyhedral oligomeric silsesquioxane (POSS) groups have been introduced into the PI film structure, so as to endow the derived films good AO resistance [22,23,24,25,26,27]. Especially, the POSS-containing groups could usually provide excellent AO resistance to the PI films while maintaining the intrinsic thermal stability due to the high contents of silicon elements and cyclic molecular structures. In 2012, Minton et al. reported the AO effects on POSS-substituted poly (pyromellitic anhydride-4,4′-oxydianiline) (PMDA-ODA) films (POSS-PI) films in low earth orbit [28]. Laboratory and spaceflight experiments have shown that POSS-PI films are highly resistant to AO attack with the erosion yields as little as 1% those of PI (PMDA-ODA) film. Although the effects of POSS components on the AO resistance of the PI films were reported in detail, their effects on the other important properties of the derived films, including thermal, mechanical, and optical properties of the films were not addressed.

In the current work, as one of our continuous work developing high-performance PI films with excellent AO resistance [29], a series of POSS-substituted PI films were prepared via an aromatic diamine, *N*-[(heptaisobutyl-POSS)propyl]-3,5-diaminobenzamide (DABA-POSS). The influence of the latent POSS substituents on the thermal, tensile, optical, and especially the AO resistant properties of the films were investigated in detail.

## 2. Materials and Methods

### 2.1. Materials

Sublimation-purified pyromellitic dianhydride (PMDA) was purchased from Shijiazhuang HOPE Chem. Co. Ltd. (Hebei, China) and then dried at 180 °C in vacuum for 24 h prior to use. 4,4′-Oxydianline (ODA) was purchased from Tokyo Chemical Industry (TCI) Co., Ltd. (Tokyo, Japan) and purified by sublimation under reduced pressure. Aminopropylisobutyl POSS was purchased from Hybrid Plastics Co. Ltd. (Hattiesburg, MS, USA) (Product code: AM0265) and used as received. 3,5-Dinitrobenzyl chloride (DNBC) was purchased from TCI (Tokyo, Japan) and used as received. *N*-methyl-2-pyrrolidone (NMP), *N,N*-dimethylacetamide (DMAc), and other solvents were obtained from Beijing Innochem Science & Technology Co., Ltd. (Beijing, China) and then purified by distillation prior to use.

### 2.2. Characterization

The inherent viscosity of the PI precursors, poly (amic acid) (PAA) was measured while using an Ubbelohde viscometer (As One Corp., Osaka, Japan) with a 0.5 g/dL NMP solution at 25 °C. The number average molecular weight (*M_n_*) and weight average molecular weight (*M_w_*) of the PAAs were measured using a gel permeation chromatography (GPC) system (Shimadzu, Kyoto, Japan) with a LC-20AD dual-plunger parallel-flow pumps (D1-LC), a SIL-20A is a total-volume injection-type auto-sampler, a CTO-20A column oven, and a RID-20A detector. HPLC grade NMP was used as the mobile phase at a flow rate of 1.0 mL/min. The attenuated total reflectance Fourier transform infrared (ATR-FTIR) spectra of the PI films were recorded on a Iraffinity-1S FT-IR spectrometer (Shimadzu, Kyoto, Japan). Nuclear magnetic resonances (^1^H-NMR) of the DABA-POSS diamine were performed on an AV 400 spectrometer (Ettlingen, Germany) operating at 400 MHz in CDCl_3_. Ultraviolet-visible (UV-Vis) spectra were recorded on a Hitachi U-3210 spectrophotometer (Tokyo, Japan) at room temperature. Wide-angle X-ray diffraction (XRD) was conducted on a Rigaku D/max-2500 X-ray diffractometer (Tokyo, Japan) with Cu-Kα1 radiation, operating at 40 kV and 200 mA. X-ray photoelectron spectroscopy (XPS) data were obtained with an ESCALab220i-XL electron spectrometer (Thermo Fisher Scientific Co. Ltd., Waltham, MA, USA) while using 300 W of MgKα radiation. The base pressure was 3 × 10^−9^ mbar. The binding energies were referenced to the C1s line at 284.8 eV from the adventitious carbon. Field emission scanning electron microscopy (FE-SEM) was carried out using a Technex Lab Tiny-SEM 1540 (Tokyo, Japan) with an accelerating voltage of 15 KV for imaging. Pt/Pd was sputtered on each film in advance of the SEM measurements.

The yellow index (YI) values of the POSS-PI films were measured using an X-rite color i7 spectrophotometer (X-Rite, Inc., Grand Rapids, MI, USA) with PI samples at a thickness of 25 μm in accordance with the procedure that is described in ASTM D1925. The color parameters were recorded according to a CIE Lab equation. *L** is the lightness, where 100 means white and 0 implies black. A positive *a** means a red color and a negative one indicates a green color. A positive *b** means a yellow color and a negative one indicates a blue color.

Thermo-gravimetric analysis (TGA) was performed on a TA-Q50 thermal analysis system (New Castle, DL, USA) at a heating rate of 20 °C/min. in nitrogen. Differential scanning calorimetry (DSC) was recorded on a TA-Q100 thermal analysis system (New Castle, DL, USA) at a heating rate of 10 °C/min. in nitrogen. Dynamic mechanical analysis (DMA) was recorded on a TA-Q800 thermal analysis system (New Castle, DL, USA) in nitrogen at a frequency of 1 Hz and heating rate of 5 °C /min. Thermomechanical analysis (TMA) was recorded on a TA-Q400 thermal analysis system (New Castle, DL, USA) in nitrogen at a heating rate of 10 °C/min.

The tensile properties were performed on an Instron 3365 Tensile Apparatus (Norwood, MA, USA) with 80 × 10 × 0.05 mm^3^ samples in accordance with GB/T 1040.3-2006 at a drawing rate of 2.0 mm/min. At least six test samples were tested for each PI film and the results were averaged.

The atomic oxygen (AO) exposure experiments were tested in a ground-based AO effects simulation facility in BISEE (Beijing Institute of Spacecraft Environment Engineering, Beijing, China). The AO beam is a mixture of ions, oxygen atoms, and other species, in proportions that have not been characterized. The mass loss of reference Kapton^®^ exposed to AO characterizes the AO flux. The facility produces an AO flux at a magnitude of 10^15^ atoms/cm^2^/s and the total AO exposure dose is 2.16 × 10 ^21^ atoms/cm^2^ in the current work. The average kinetic energy of AO beam falls in the range of 3~8 eV. The AO exposure was performed on square POSS-PI film samples with the size of 20 (length) × 20 (width) × 0.05 (thickness) mm^3^. The films were exposed to AO at a fluence of 2.1 × 10^2^^1^ atoms/cm^2^ and the mass loss was determined. The erosion yield of the sample, *E_s_*, is calculated through the following Equation (1) [30]:(1)$$Es=\frac{\Delta M_{s}}{A_{s}\rho_{s}F}$$
where, *E_s_* = erosion yield of the sample (cm^3^/atom); Δ*M_s_* = mass loss of the sample (g); *A_s_* = surface area of the sample exposed to atomic oxygen attack (cm^2^); *ρ_s_* = density of the sample (g/cm^3^); and, *F* = AO fluence (atoms/cm^2^). Because Kapton film has a well characterized erosion yield, which is 3.0 × 10^−24^ cm^3^/atom, and all of the present PI samples are supposed to possess similar densities and exposed area with Kapton in the AO attacking experiments, and the *E_s_* of the PIs can therefore be calculated while using a simplified Equation (2):(2)$$E_{s}=\frac{\Delta M_{s}}{\Delta M_{Kapton}}E_{Kapton}$$
where, *E_Kapton_* stands for the erosion yield of Kapton standard, which is 3.0 × 10^−24^ cm^3^/atom; Δ*M_Kapton_* stands for the mass loss of Kapton standard.

### 2.3. Monomer Synthesis

*N*-[(heptaisobutyl-POSS)propyl]-3,5-diaminobenzamide (DABA-POSS) was synthesized in our laboratory according to a modified procedure that is reported in the literature [28]. The hydrogenation of the dinitro compound, *N*-[(heptaisobutyl-POSS)propyl]-3,5-dinitrobenzamide (DNBA-POSS), was performed under the reduction of hydrazine monohydrate with the palladium on activated carbon (Pd/C) catalyst instead of the platinum oxide/hydrogen (PtO_2_/H_2_) system.

Yield: 93.2%. Purity: 99.5% (liquid chromatography analysis). Melting point: 202.5 °C (DSC peak temperature). FT-IR (KBr, cm^−1^): 3444, 3356, 2955, 2872, 1626, 1597, 1527, 1466, 1365, 1333, 1230, 1111, 839, and 742. ^1^H-NMR (300 MHz, DMSO-*d*_6_, ppm): 6.45 (s, 2H), 6.12 (s, 1H), 5.96 (s, 1H), 3.67 (s, 4H), 3.41–3.39 (q, 2H), 1.85 (m, 7H), 1.67 (m, 2H), 0.97 (m, 42H), and 0.68-0.61 (*m*, 16H). Elemental analysis: C_38_H_77_N_3_O_13_Si_8_: Cald. C, 45.25%, H, 7.69%, N, 4.17%; Found: C, 45.01%, H, 7.71%, and N, 4.26%.

### 2.4. PAA Synthesis and PI Film Preparation

A series of PAA solutions with different contents of DABA-POSS were prepared. The representative synthesis procedure could be illustrated by the preparation of PAA-20. Ultra-dry DMAc (200.0 g) was added into a 500 mL three-necked flask equipped with a mechanical stirrer, a cold-water bath, and a nitrogen inlet and the reaction system was filled with the continuous nitrogen flow. Subsequently, ODA (33.0920 g, 165.30 mmol) and DABA-POSS (18.272 g, 18.11 mmol) were added and the reaction system was cooled to −5–0 °C. The diamine solution with the deep pink color was obtained after stirring for 10 min. under the flow of nitrogen. Subsequently, PMDA (40.000 g, 183.40 mmol) was added to the solution together with an additional DMAc (164.0 g). The solid content of the reaction system was controlled to be 20 wt%. The reaction mixture was stirred for 1h and then the cold-water bath was removed. The reaction was prolonged for another 20 h at room temperature. Afterwards, the viscous brown-yellow solution was obtained, which was purified by filtration through a 2.0 μm polytetrafluoroethylene (PTFE) filter in order to afford the PAA-20 solution.

The purified PAA-20 solution was cast on a clean glass substrate with a scraper blade. The thickness of the wet PAA-20 film was adjusted by regulating the slit height of the scraper blade. Subsequently, the PI-20 films with different controlled thicknesses were obtained by thermally baking the PAA-20 solution in an oven with nitrogen gas flow, according to the following procedure: 80 °C/3 h, 150 °C/1 h, 180 °C/1 h, 250 °C/1 h, and 300 °C/1 h.

The other PAA solutions and the corresponding PI films, including PI-10, PI-15, PI-25, and PI-30 films, were prepared according to a similar procedure mentioned above. The PI-0 (PMDA-ODA) film without POSS components was also prepared for comparison.

## 3. Results and Discussion

### 3.1. Monomer Synthesis

Figure 1 shows the diamine monomer, DABA-POSS, with latent heptaisobutyl-substituted POSS in the side chain was synthesized via a two-step procedure. First, the dinitro compound, *N*-[(-heptaisobutyl-POSS)propyl]-3,5-dinitrobenzamide (DNBA-POSS), was prepared from the starting DNBC and aminopropylisobutyl POSS (AM0265) with dichloromethane as the solvent and triethylamine as the acid absorbent. The reaction was performed at the temperature below 0 °C in order to avoid the occurrence of side reactions. Subsequently, the dinitro compounds were catalytically hydrogenated with hydrazine monohydrate to afford the target DABA-POSS compound. The total yield was about 93.2%. The highly pure DABA-POSS diamine was obtained as colorless crystals with the sharp endothermic melting peak at 202.5 °C in the DSC measurement. The light-sensitive DABA-POSS diamine was easily to be oxidized in air and the color turned from colorlessness to pinkness during storage.

The chemical structure of the DABA-POSS diamine was identified the FT-IR, ^1^H-NMR, and elemental analysis measurements. Figure 2 depicts the ^1^H-NMR spectra of DABA-POSS diamine together with its dinitro precursor. It could be clearly observed that the protons in the POSS side chains (H_5_~H_10_) revealed absorptions at the farthest upfield region in the spectra. On the country, the protons in the benzene units (H_1_, H_2_) showed absorptions at the farthest downfield region in the spectra, although the chemical shifts were not the same for DABA-POSS and DNBA-POSS. Proton H_7_ that was attached with the silicon elements exhibited the signals with the lowest chemical shift values in both of the spectra for DABA-POSS and DNBA-POSS. For DABA-POSS, clear absorption for the amino hydrogen protons was detected at 3.67 ppm, as in Figure 2b. The information is in good agreement with the anticipated chemical structures for the target compounds [28]. In addition, the FT-IR and elemental analysis results also supported the successful preparation for the diamine.

### 3.2. PAA Synthesis and Film Preparation

A series of PAA varnishes, a total of six samples, including one system without POSS component, PI-0 (PMDA-ODA) and five copolymers with different DABA-POSS contents, PI-10, PI-15, PI-20, PI-25, and PI-30 (numbers in the codes indicate the percent weight ratio of DABA-POSS in the total weights of the derived PI films), were prepared, respectively, with the procedure that is shown in Figure 3. The DABA-POSS diamine exhibited inferior solubility in DMAc than that of ODA, which might be due to the high molecular weight of the diamine or the non-polar isobutyl groups in the diamine. After long-time polymerization, DABA-POSS was totally reacted and afforded PAA varnishes with high molecular weights. Table 1 lists the inherent viscosities ([*η*]_inh_) and molecular weights, including the number average molecular weight (*M*_n_), weight average molecular weight (*M*_w_), and the polydispersity index (PDI) of the PAA varnishes. The [*η*]_inh_ and molecular weights of the PAA varnishes gradually decreased with the increasing contents of the POSS diamine in the systems. For example, PAA-30, with the highest POSS diamine content in the systems, exhibited the [*η*]_inh_ and *M*_n_ values of 0.93 dL/g and 7.43 × 10^4^ g/mol, respectively, which are apparently lower than those of the pristine PAA-0 without POSS component ([*η*]_inh_ = 1.33 dL/g; *M*_n_ = 10.70 × 10^4^ g/mol). This decreasing trend in the molecular weights of the PAA varnishes is mainly attributed to the relatively lower polymerization reactivity of the POSS diamine when compared to that of ODA. Nevertheless, the current level of the molecular weights of the PAA varnishes could guarantee the film-forming ability and good mechanical properties of the derived PI films.

A series of PI films were prepared from the corresponding PAA varnishes according to the procedure that is shown in Figure 4. The PAA varnishes were cast onto the clean glass substrates and cured at elevated temperatures from 80 to 300 °C under the protection of nitrogen gas. Flexible and tough free-standing PI films with controlled thickness were obtained, whose chemical structures were confirmed and various properties were evaluated.

Figure 5 shows the ATR-FTIR spectra of the PI films, together with the assignments of representative absorptions. First, the characteristic absorptions of the imide rings, including the absorptions at 1778 cm^−1^ that were assigned to the asymmetrical carbonyl stretching vibrations, at 1723 cm^−1^ assigned to the symmetrical carbonyl stretching vibrations, at 1373 cm^−1^ for the C–N stretching vibrations, and at 724 cm^−1^ for the carbonyl out-of-plane bending vibrations were all clearly observed. The characteristic absorptions of the saturated C-H bonds at 2955 cm^−1^ and the Si-O-Si stretching vibrations at 1130 cm^−1^ were only observed in the spectra of PI-10~PI-30, which indicated the successful introduction of the POSS units in the PI films.

Although the POSS components were successfully introduced into the molecular structures of the PI films via copolymerization instead of external adding as fillers, obvious phase separation was still observed with the increasing contents of DABA-POSS. This could be indirectly confirmed by the change of the optical parameters of the films. Figure 6 shows the two-dimensional (2D) and three-dimensional (3D) maps of CIE Lab optical parameters of the PI films, respectively. Apparently, the yellow indices (*b*^*^) and haze values of the PI films increased with the increasing contents of POSS units. For example, the PI-30 film showed the *b*^*^ and haze values of 96.42 and 31.70%, respectively, which are obviously higher than those of the pristine PI-0 film (*b*^*^ = 84.23; haze = 0.68). It is the phase separation instead of the crystallization of the molecular chains that caused the opaque appearance of the PI films with high POSS units. This could be confirmed by the XRD measurement that is shown in Figure 7. All of the PI films exhibited amorphous nature. This is mainly due to the bulky POSS side chains and flexible ether linkages in the PI molecular chains, which prohibit the condense packing of the molecular chains.

### 3.3. Thermal and Mechanical Properties

The effects of the POSS units on the thermal properties and high-temperature dimensional stability of the PI films were investigated by TGA and TMA, respectively. Figure 8 shows the TGA curves of the PI films and the thermal data are listed in Table 2. Basically, all of the PI films exhibited good stability up to 500 °C in nitrogen, after which they began to decompose and revealed the 5% weight loss temperatures (*T*_5%_) in the range of 512–574 °C. At the end of the TGA measurement, the PI films left approximately 57–60% of their original weights at 750 °C. The incorporation of the POSS units into the PI films decreased the initial thermal decomposition temperatures of the PI films; however, it slightly increased the values of residual weight ratio at 750 °C (*R*_w750_). For instance, the PI-30 film showed the *T*_5%_ and *R*_w750_ values of 506 °C and 59.3%, respectively. The former value is 68 °C lower than that of the pristine PI-0 film (*T*_5%_ = 574 °C) and the latter value is slightly higher than that of the pristine PI-0 film (*R*_w750_ = 57.4%). The decrease of the *T*_5%_ values of the POSS-PI films is mainly due to the thermal unstable nature of the isobutyl groups in the POSS units, while the slight increase of the *R*_w750_ values might be attributed to the formation of heat-resistant silicon oxide at elevated temperatures, due to the oxidation reaction of the silicone elements with the trace amount of oxygen in the gas.

Subsequently, the high-temperature dimensional stability of the PI films was investigated by the TMA measurements and the CTE values of the PI films in the temperature range of 50–250 °C were recorded in the TMA plots that are shown in Figure 9 and Table 2. It can be seen from the thermal expansion behaviors of the PI films that the incorporation of POSS units sacrificed the dimensional stability of the PI films at elevated temperatures. The PI-30 film showed the CTE value of 54.6 × 10^−6^/K, which is obviously higher than that of the pristine PI-0 film (CTE = 28.9 × 10^−6^/K). This might be due to the internal plasticization effects of the latent POSS units in the PIs. In addition, all the POSS-PI films did not show clear glass transitions in the differential scanning calorimetry (DSC) measurement in the temperature range of 30 to 400 °C. In the TMA measurement, PI-0 showed clear glass transition in the temperature range of 300 to 350 °C. However, the POSS-PI films did not show an obvious glass transition before 320 °C. This phenomenon indicates that the incorporation of POSS units might delay the softening of the PI films; that is increasing the glass transition temperatures (*T*_g_) of the PI films, although the CTE values increased at the same time. This result agreed well with the POSS-containing PI films that are reported in the literature [31].

The influence of the POSS units on the tensile properties of the PI films was investigated and the tensile data are summarized in Table 2. Generally speaking, the tensile properties are very important for the reliability of PI films in practical applications, especially for applications such as spacecrafts that are not easy to repair materials. PI films are generally used as thermal protection materials in spacecrafts, which usually does not bear large loadings. However, high strength, high modulus, high elongation at breaks, and good tear resistance of the PI films are often required. It can be deduced from the tensile data that the incorporation of POSS units apparently deteriorated the tensile properties of the PI films. All of the tensile strength (*T*_S_), tensile modulus (*T*_M_), and elongations at breaks (*E*_b_) values of the PI films decreased with the increasing contents of the POSS units in the polymers. For example, the PI-30 film showed the *T*_S_, *T*_M_, and *E*_b_ values of 75.0 MPa, 1.55 GPa, and 16.1%, respectively. These values are obviously lower than those of the PI-0 film (*T*_S_ = 131.0 MPa; *T*_M_ = 1.90 GPa; and *E*_b_ = 73.2%). This result is in good consistence with the analogous POSS-containing PI films that are reported in the literature [32]. The deterioration of the tensile properties of the PI films is, on one hand, due to the intrinsic molecular structures, such as low contents of flexible linkages in the POSS-PI films, and, on the other hand, due to the decreased molecular weights of the PI films.

### 3.4. Atomic Oxygen Resistant Properties

The AO erosion behaviors of the current PI films were investigated in the ground-simulated facility with the total AO dose of 2.16 × 10^21^ atoms/cm^2^. The AO-resistant ability of the PI films was justified by the weight loss during the AO exposure, which was recorded in Figure 10 and in Table 3. The erosion yields (*E_s_*) of the PI films were calculated according to equation (2) in Section 2.2 with a hypothesis that the POSS-PI films had the same density with the Kapton^®^ reference. It can be clearly observed that the incorporation of POSS components obviously increased the AO resistance of the PI films. All of the POSS-PI films survived from the AO exposure. The *E_s_* values of the PI films decreased with the increasing contents of the POSS components in the films. The PI-30 film showed an *E_s_* value of 1.1 × 10^−25^ cm^3^/atom, which was approximately 3.67% of the Kapton^®^ film (*E_s_* = 3.0 × 10^−24^ cm^3^/atom).

In order to reveal the mechanism for the enhancement of the AO resistance of the PI films via incorporation of POSS units, the surface chemical compositions and the micro-morphologies of the films were investigated, respectively. First, Figure 11 and Table 4 compared the XPS results of the PI films before and after AO exposure. Apparently, after AO exposure, the relative atomic concentrations of silicon (Si) and oxygen (O) obviously increased, while those of the carbon (C) and nitrogen (N) decreased sharply. This result indicated the formation of inert silica or silicate passivation layers on the surface of the POSS-PI films after AO exposure, which efficiently prevented the further erosion of the under-layer materials.

Figure 12 shows the micro-morphologies of the PI films that were detected by SEM measurements after AO exposure. it can be seen that, with the equal magnification (×8000) and scale (5 μm), the passivation layer formed on the surface of the PI films after AO exposure gradually tended to be compact with the increasing of the POSS contents in the PI films. With much higher magnification (×250,000), we can clearly observe the granular particles on the surface of the PI films, which are composed the passivation layers. Because the formation of the passivation layer required a period of in-situ reaction time between the PI film and AO, a relatively large mass loss was observed at the initial stage of AO exposure, as shown in Figure 10. As the passivation layer became increasingly dense, the mass loss of the PI films tended to be nonlinear, especially for the samples with higher POSS contents. However, because the AO erosion experiment in this study was carried out under static conditions, it is necessary to further investigate whether the passivation layer could form close adhesion with the under-layer film when the films were impacted by external forces in practical application.

## 4. Conclusions

A series of POSS-containing PI films were prepared in order to enhance the AO resistance of the common PI (PMDA-ODA) film. The purpose was successfully achieved, although the incorporation of POSS units deteriorated the thermal and tensile properties of the PI films to some extent. Overall, it is more optimal to control the proportion of POSS diamine unit in the PI film at 20 wt%. The derived PI-20 film possessed the best combined properties in the series, including the *T*_5%_ of 526 °C, the CTE of 56.1 × 10^−6^/K, the *T*_S_ of 97.6 MPa, the *T*_M_ of 1.69 GPa, the *E*_b_ of 28.0%, and AO erosion yield of 1.7 × 10^−25^ cm^3^/atom. For such series of POSS-containing PI films, the relatively high CTE and low tensile modulus might be the weakness for future practical applications. How to enhance the high-temperature dimensional stability (CTE ≤ 40 ×10^−6^/K) and modulus (*T*_M_ ≥ 3.0 GPa) of these PI films will be the main challenges to face in the future research work. Related research is now under investigation in our laboratory.

## Figures and Tables

**Figure 1 polymers-12-02865-f001:**
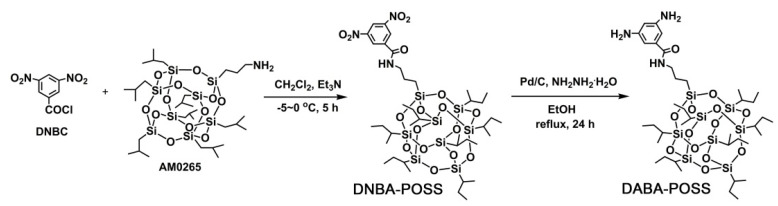
Synthesis of *N*-[(heptaisobutyl-POSS)propyl]-3,5-diaminobenzamide (DABA-POSS) diamine.

**Figure 2 polymers-12-02865-f002:**
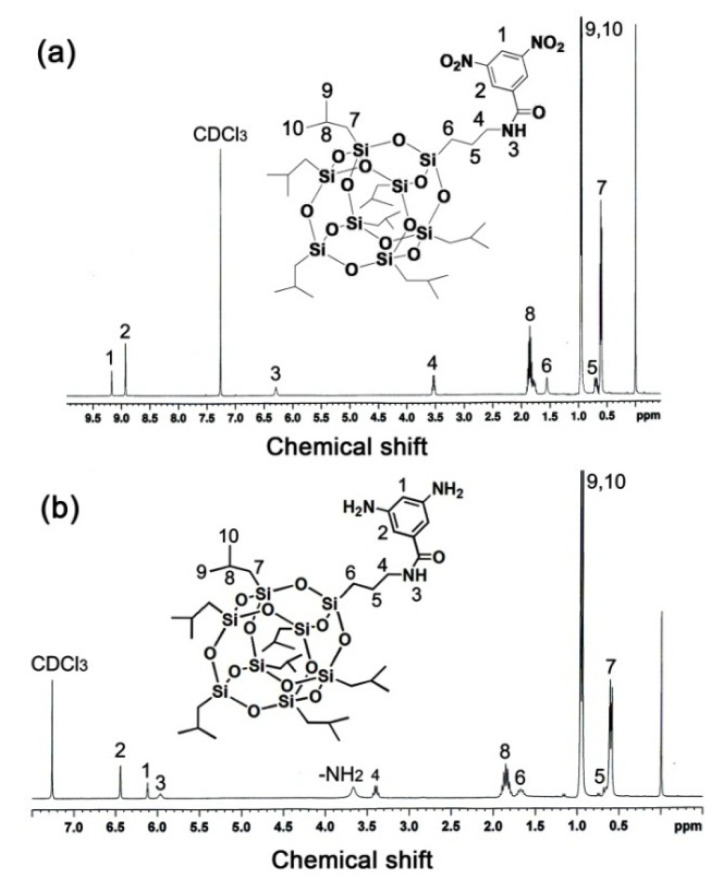
^1^H-NMR spectra of the chemical compounds. (**a**) *N*-[(heptaisobutyl-POSS)propyl]-3,5-dinitrobenzamide (DNBA-POSS); and, (**b**) DABA-POSS.

**Figure 3 polymers-12-02865-f003:**
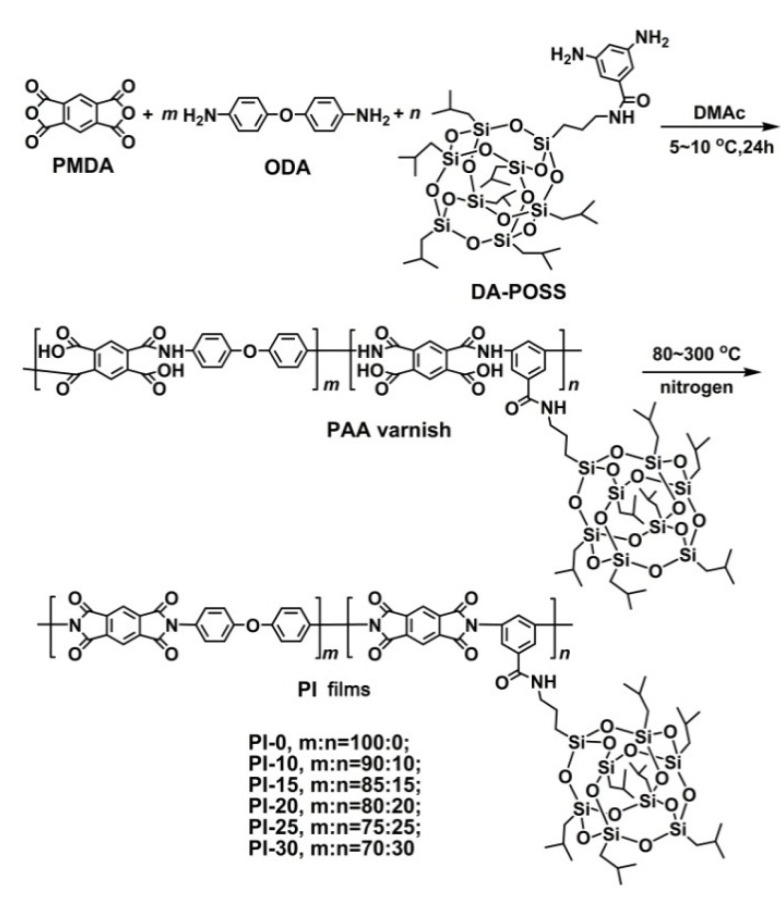
Synthesis of polyimides (PIs) and PI-ref films.

**Figure 4 polymers-12-02865-f004:**
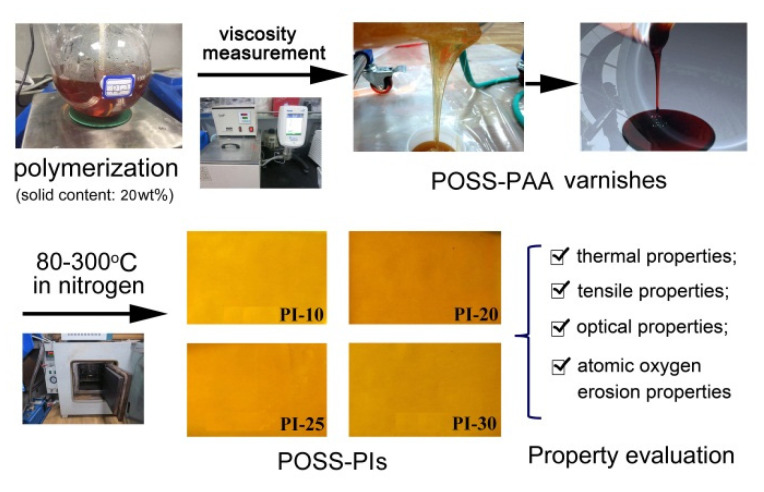
Preparation procedure of PI films.

**Figure 5 polymers-12-02865-f005:**
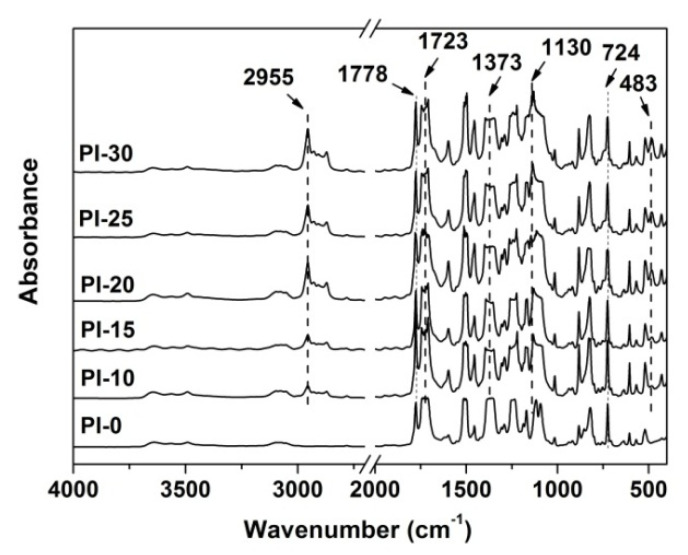
Attenuated total reflectance Fourier transform infrared (ATR-FTIR) spectra of PI coatings.

**Figure 6 polymers-12-02865-f006:**
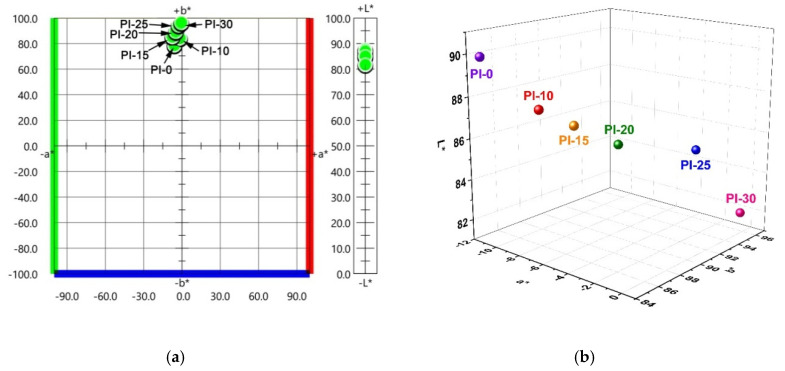
CIE Lab color parameters of POSS-substituted poly (pyromellitic anhydride-4,4′-oxydianiline) (PMDA-ODA) films (POSS-PI) films. (**a**) Two-dimensional (2D) map; and, (**b**) three-dimensional (3D) map.

**Figure 7 polymers-12-02865-f007:**
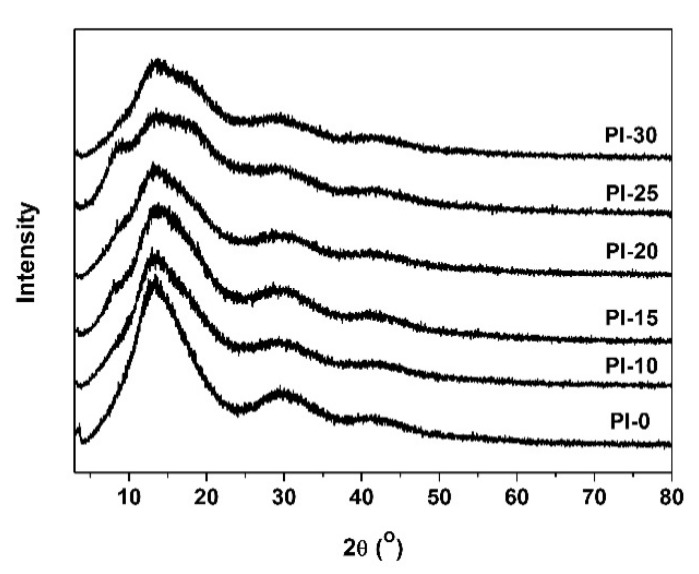
X-ray diffraction (XRD) patterns of POSS-PI films.

**Figure 8 polymers-12-02865-f008:**
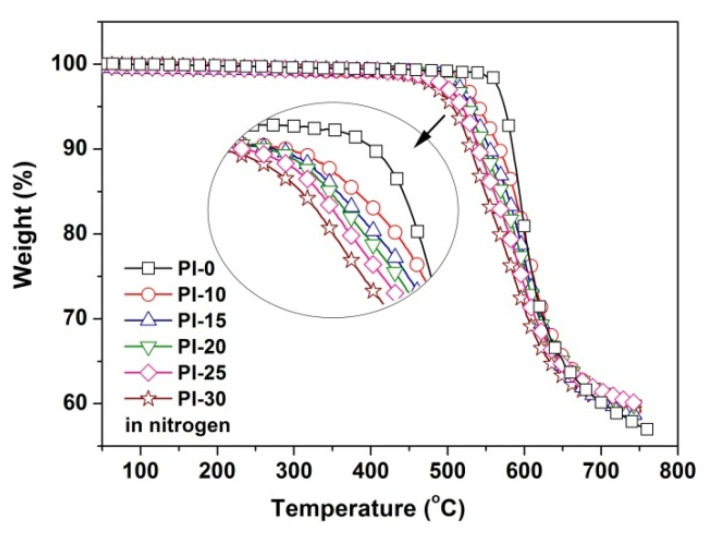
Thermo-gravimetric analysis (TGA) plots of PI films in nitrogen.

**Figure 9 polymers-12-02865-f009:**
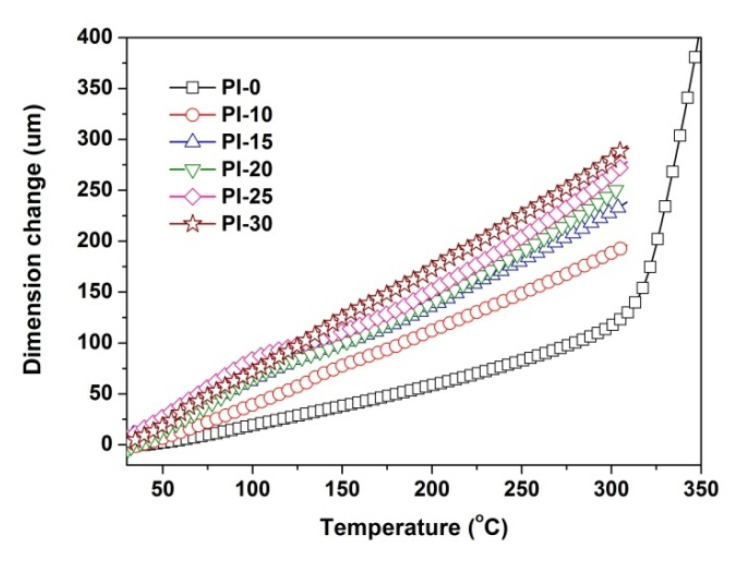
TMA curves of POSS-PI films.

**Figure 10 polymers-12-02865-f010:**
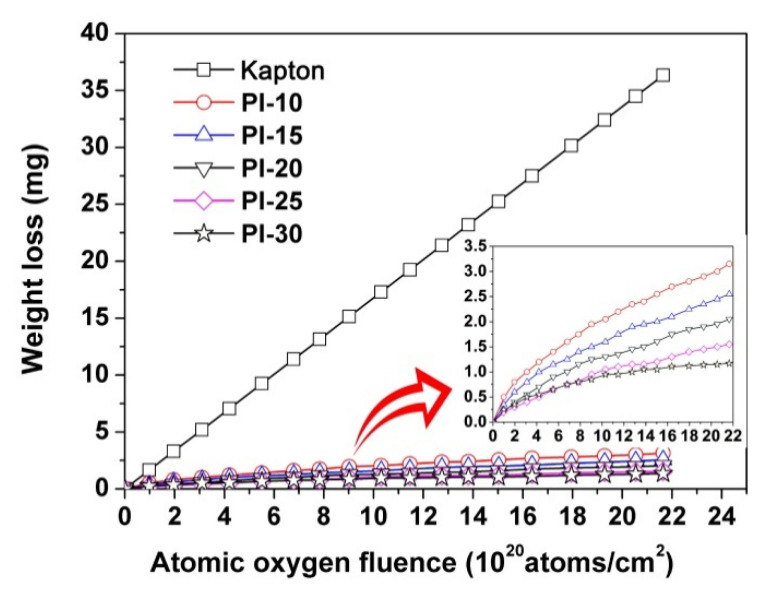
Weight loss of POSS-PI films during atomic-oxygen (AO) erosion (AO fluence: 2.16 × 10^21^ atoms/cm^2^).

**Figure 11 polymers-12-02865-f011:**
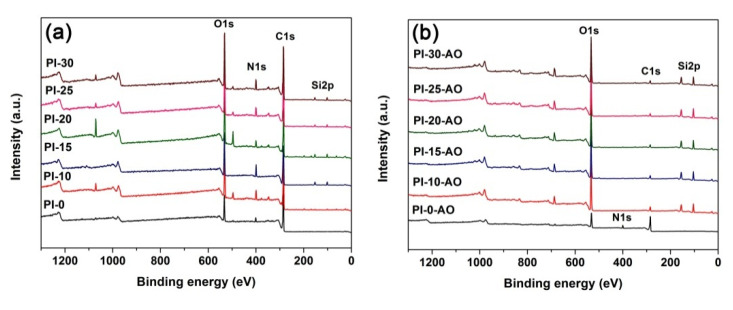
XPS spectra of PI films. (**a**) pristine samples; (**b**) AO-exposed samples.

**Figure 12 polymers-12-02865-f012:**
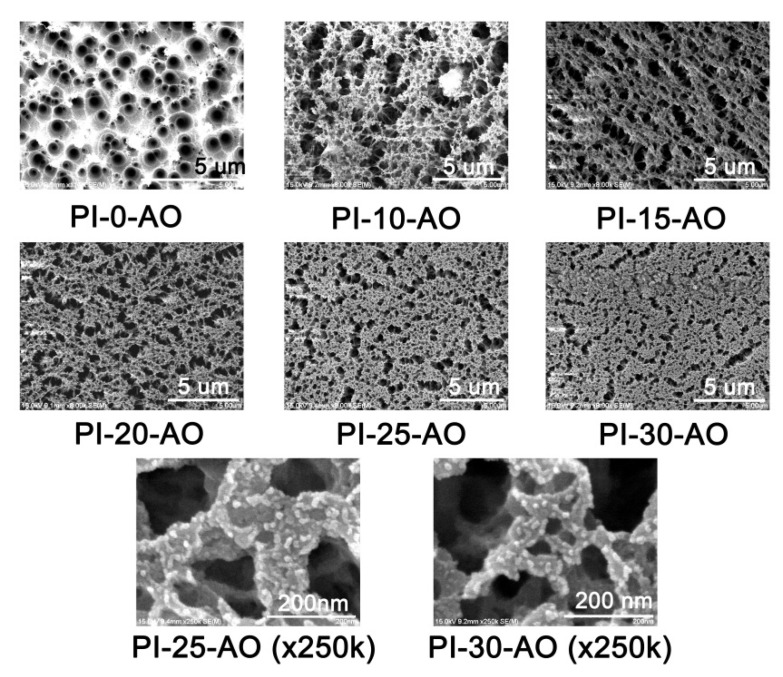
SEM micrographs of POSS-PI films after AO erosion (AO fluence: 2.16 × 10^21^ atoms/cm^2^).

**Table 1 polymers-12-02865-t001:** Inherent viscosities and molecular weights of poly (amic acid)s (PAAs).

PAA	[*η*]_inh_ ^a^ (dL/g)	*M*_n_^b^ (×10^4^ g/mol)	*M*_w_^b^ (×10^4^ g/mol)	PDI ^b^
PAA-0	1.33	10.70	17.11	1.59
PAA-10	1.28	9.87	15.88	1.61
PAA-15	1.20	9.61	16.29	1.69
PAA-20	1.13	9.57	15.60	1.63
PAA-25	1.04	8.56	14.34	1.68
PAA-30	0.93	7.43	13.63	1.83

^a^ Inherent viscosities measured with a 0.5 g/dL PAA solution in NMP at 25 °C; ^b^
*M*_n_: number average molecular weight; *M*_w_: weight average molecular weight; PDI: polydispersity index, PDI = *M*_w_/*M*_n_.

**Table 2 polymers-12-02865-t002:** Optical, thermal, and tensile properties of PI films.

PI	CIE Lab Parameters ^a^				Thermal Properties ^b^			Tensile Properties ^c^		
	*L**	*a**	*b**	haze (%)	*T*_5__%_ (^o^C)	*R*_w7__5__0_ (%)	CTE (×10^−6^/K)	*T*_S_ (MPa)	*T*_M_ (GPa)	*E*_b_ (%)
PI-0	89.93	−11.09	84.23	0.68	574	57.4	28.9	131.0	1.90	73.2
PI-10	87.76	−7.48	85.41	4.78	534	59.1	45.6	111.7	1.87	29.4
PI-15	87.05	−5.86	86.64	7.65	528	58.5	50.4	108.1	1.77	28.9
PI-20	85.84	−4.87	89.46	9.55	524	59.6	56.1	97.6	1.69	28.0
PI-25	85.05	−2.66	94.41	14.53	519	59.9	55.0	93.3	1.64	25.6
PI-30	81.79	−0.49	96.42	31.70	512	59.3	54.6	75.0	1.55	16.1

^a^*L**, *a**, *b**, see 2.2 Characterizations part. ^b^*T*_5%_: Temperatures at 5% weight loss; *R*_w700_: Residual weight ratio at 750 ^o^C in nitrogen; CTE: linear coefficient of thermal expansion in the range of 50–250 °C. ^c^*T*_S_: Tensile strength; *T*_M_: Tensile modulus; *E*_b_: Elongation at break.

**Table 3 polymers-12-02865-t003:** AO effects and erosion yields for POSS-PI films.

Sample	*W*_1_^a^ (mg)	*W*_2_^a^ (mg)	Δ*W* ^a^ (mg)	*E_s_*^b^ (10^−25^ cm^3^/atom)
PI-0 (Kapton)	4535	900	3635	30.0
PI-10	2775	2460	315	2.6
PI-15	2225	1970	255	2.1
PI-20	2335	2130	205	1.7
PI-25	2670	2515	155	1.3
PI-30	2980	2845	135	1.1

^a^*W*_1_: Weight of the sample before irradiation; *W*_2_: Weight of the sample before irradiation; Δ*W*: Weight loss of the sample during irradiation, Δ*W* = *W*_1_ − *W*_2_; ^b^ Erosion yield.

**Table 4 polymers-12-02865-t004:** XPS results for the unexposed and exposed POSS-PI films.

Samples	Unexposed Samples				AO Exposed Samples			
	Si2p	C1s	O1s	N1s	Si2p	C1s	O1s	N1s
PI-10	3.92	67.92	21.07	6.86	24.27	12.04	53.52	0.41
PI-15	3.23	65.27	22.95	6.68	24.30	10.24	54.32	0.43
PI-20	3.63	59.95	25.55	4.25	24.50	8.86	54.76	0.35
PI-25	2.29	67.68	21.87	5.67	24.29	8.54	55.02	0.34
PI-30	3.08	64.63	23.63	6.14	23.42	9.51	53.71	0.31

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
