# Peer review of "Preparation and Properties of Intrinsically Atomic-Oxygen Resistant Polyimide Films Containing Polyhedral Oligomeric Silsesquioxane (POSS) in the Side Chains"

_polymers, 2020, doi:10.3390/polym12122865_

Round 1
Reviewer 1 Report
Dear authors,
I am herewith sending the comments about polymers 1018335 as attached file.
Thank you.

Author Response
1. Question: Incorporation of POSS components increase the AO resistance of the PI films. Why did authors not prepare polyimide having higher contents of POSS? Would you explain this reason?
Answer: Incorporation of POSS components could increase the AO resistance of the derived PI films. However, the tensile properties of the PI film gradually deteriorated with the increasing contents of the POSS diamine in the systems, as evidenced in Table 2. As we know, good tensile properties are also quite important for the practical applications of the PI films in space. Thus, we have to achieve the balance between the AO resistance and tensile properties of the developed PI films. Thus, we did not prepare PI films with higher contents of POSS in the current work.
2. Questions: CTE of PI-20 is the largest in POSS-PI films (Table 2). Why is CTE of PI-2 larger than those of PI-25 and PI-30? Would you explain this reason?
Answer: Indeed, according to our measurements, the CTE value of PI-20 (56.1×10-6/K) is much higher than those of the PI films with lower POSS contents (28.9×10-6/K, 45.6×10-6/K, and 50.4×10-6/K for PI-0, PI-10, PI-15, respectively), however a bit higher than those of the PI films with higher POSS contents (55.0×10-6/K and 54.6×10-6/K for PI-25 and PI-30, respectively). This trend might be due to the influence of the POSS components in the PI films on their dimensional stability at elevated temperatures. The bulky POSS substituents might play a role of “internal plasticizer” in the PI films. When the content of POSS components was lower than 20 wt%, the effect of this plasticization effects on the dimensional stability was obvious and the CTE values of the PI films increased apparently with the increasing contents of the POSS (from 28.9×10-6/K to 56.1×10-6/K). However, when the contents of the bulky POSS substituents were higher than 20 wt%, this plasticization effect became weak due to the increasing packing density of the bulky POSS substituents. Thus, the change degree of the CTE value reduced (from 56.1×10-6/K to 54.6×10-6/K). Therefore, we believe that when the POSS content in the PI film reaches a certain degree, the CTE values of the film tend to change little.
3. Question: Weight loss of Kapton is much larger than those of POSS-PI films (Figure 10). However, loss of PI-10-AO is not felt to be larger than that of PI-0-AO (figure 12). Would you explain this reason?
Answer: In the current research, PI-0 (PMDA-ODA) has the similar structure like Kapton. In our measurements, the weight loss of Kapton is much larger than those of the POSS-PI films according to the data shown in Figure 10. In Figure 12, the micro-morphologies of the PI films after AO erosion were shown. From the SEM images in figure 12, we cannot judge the weight loss of the PI films after AO exposure. According to the weight loss data in Table 3, the weight loss of PI-10 was 315 mg, while the PI-0 (Kapton) was 3635 mg. The former is much lower than the latter.

Reviewer 2 Report
The authors have successfully achieved to prepare the POSS-containing polyimides and investigated the properties including atomic-oxygen resistance. The present article includes some significant findings and sufficiently advances the knowledge of this area. However there are some queries and comments as follows;
- Lines #45 and the others; in the word poly(pyromellitic anhydride-co-4,4’-oxydianiline), “co” should be deleted.
- Line #48; “-296” should be corrected.
- Lines #239 and #249; PAA “resins” should be changed to PAA “varnishes”.
- Line 252; “black” should be deleted.
- Line 254; PI solution -> PAA solution, NMP -> DMAc (?)
- CIE Lab color parameters should be displayed using 3D map.
- Line #178; DABA-POSS diamine is not vaporized and it seems impossible to determine the purity by gas chromatography analysis.
I recommend this manuscript to be published in Polymers after a minor revision.
Author Response
1. Question: Lines #45 and the others; in the word poly(pyromellitic anhydride-co-4,4’-oxydianiline), “co” should be deleted.
Answer: We deleted all the “co” in our revised manuscript.
2. Question: Line #48; “-296” should be corrected.
Answer: We changed “-296” as “-269” in our revised manuscript.
3. Question: Lines #239 and #249; PAA “resins” should be changed to PAA “varnishes”.
Answer: We modified all the “PAA resins” as “PAA varnishes” in our revised manuscript.
4. Question: Line 252; “black” should be deleted.
Answer: We deleted the “black” in our revised manuscript.
5. Question: Line 254; PI solution -> PAA solution, NMP -> DMAc (?)
Answer: We changed “PI solution” as “PAA solution” in our revised manuscript.
6. Questions: CIE Lab color parameters should be displayed using 3D map.
Answer: Thanks for the suggestion, we added the 3D map for the CIE Lab figure (figure 6b) in our revised manuscript.
7. Questions: Line #178; DABA-POSS diamine is not vaporized and it seems impossible to determine the purity by gas chromatography analysis.
Answer: We changed “gas chromatography analysis” as “liquid chromatography analysis” in our revised manuscript.
